# Combining multiplexed assays of variant effect for enhanced *BRCA2* variant classification

Chunling Hu [1,14], Sounak Sahu [2,13,14], Wenan Chen [3], Melissa Galloux [4], Marcy E. Richardson [5], Megan F. Bishop[1], Rachid Karam [5], Tina Pesaran [5], Jie Na[3], Huaizhi Huang [1,6,7], Jeffrey N. Weitzel [8], Katherine N. Nathanson [9], Siddhartha Yadav [10], Nicholas J. Boddicker[3], Susan M. Domchek [9], Alvaro N. Monteiro [11], Edwin S. Iversen [12], Shyam K. Sharan [2] ✉ & Fergus J. Couch [1,3] ✉

Determining the clinical relevance of *BRCA2* variants of uncertain significance is critical for informed risk management. Recently, two saturation genome editing studies assessed the functional effects of all single nucleotide variants in the BRCA2 C-terminal DNA Binding Domain. To improve the accuracy of functional data used for ACMG/AMP variant classification, we combined results from these studies in four composite models and evaluated the performance of each model using variants with known classifications. Here, we show that an "Integrated VarCall Model", which combined raw functional data for 6383 variants from the original studies, yielded 98.8% accuracy and outperformed the original studies and other combined data models. Incorporation of the "Integrated VarCall Model" functional data with other sources of evidence according to ClinGen *BRCA1/2* variant curation expert panel specifications resulted in classification of 5926 (92.8%) *BRCA2* variants as pathogenic (*n* = 735) or benign (*n* = 5191) and provides valuable insights for individuals with *BRCA2* variants.

Loss-of-function germline pathogenic variants (PVs) in *BRCA2* have been associated with a 69% lifetime risk of developing breast cancer and a 15% risk of developing ovarian cancer[1–3]. Carriers of germline PVs can benefit from enhanced mammography and MRI screening, options for risk reducing salpingo-oophorectomy and mastectomy, awareness of increased risks of second breast cancers, ovarian, pancreatic and prostate cancer, risk assessment for family members, and cancer treatment with targeted poly ADP ribose polymerase inhibitors (PARPi). However, genetic testing has also identified many variants of uncertain significance (VUS) in *BRCA2*, that do not qualify patients for these management strategies. The lack of knowledge about the benign or pathogenic nature of these variants increases uncertainty in patient management and psychological burden for VUS carriers and their families. Thus, there is an

[1]Department of Laboratory Medicine and Pathology, Mayo Clinic, Rochester, MN, USA. [2]Mouse Cancer Genetics Program, Center for Cancer Research, National Cancer Institute, Frederick, MD, USA. [3]Department of Quantitative Health Sciences, Mayo Clinic, Rochester, MN, USA. [4]Independent Bioinformatician, Marseille, France. [5]Ambry Genetics, Aliso Viejo, CA, USA. [6]Department of Molecular Pharmacology and Experimental Therapeutics, Mayo Clinic, Rochester, MN, USA. [7]Graduate School of Biomedical Sciences, Mayo Clinic, Rochester, MN, USA. [8]Division of Precision Prevention, University of Kansas Medical Center, Kansas City, KS, USA. [9]Perelman School of Medicine at the University of Pennsylvania, Philadelphia, PA, USA. [10]Department of Oncology, Mayo Clinic, Rochester, MN, USA. [11]Department of Epidemiology, H. Lee Moffitt Cancer Center,, Tampa, FL, USA. [12]Department of Statistical Science, Duke University, Raleigh Durham, NC, USA. [13]Present address: Department of Cell Biology, NYU Grossman School of Medicine, New York, NY, USA. [14]These authors contributed equally: Chunling Hu, Sounak Sahu. ✉e-mail: sharans@mail.nih.gov; couch.fergus@mayo.edu

urgent need for formal classification of VUS in *BRCA2* as either pathogenic or benign.

The ClinGen Variant Curation Expert Panel (VCEP) for *BRCA1/2* specifications for *BRCA2* variant classification based on American College of Medical Genetics (ACMG) and the Association for Molecular Pathology (AMP) classification guidelines have been developed and applied to a subset of *BRCA2* variants[4–7]. These expert panel-based classifications are listed on the ClinVar website along with proposed classifications of variants by genetic testing laboratories. Overall ClinVar lists over 5000 *BRCA2* VUS and variants with conflicting interpretations.

Among all missense variants, those with established pathogenic status are located in the C-terminal DNA binding domain (DBD) of BRCA2 (encoded by exons 15–26). On this basis, two CRISPR-Cas9 saturation genome editing (SGE) studies were carried out to evaluate the impact of all possible single-nucleotide variants (SNVs) in *BRCA2* exons 15-26 on cell survival of the haploid human HAP1 cell line (Huang et al., 2025)[8] and a humanized mouse embryonic stem (ES) cell line (Sahu et al., 2025)[9]. The Sahu et al.[9] study employed a two-component Gaussian Mixture Model (GMM), whereas Huang et al.[8] utilized Var-Call, a Bayesian hierarchical model that embeds a GMM model with position and batch effect adjustment[8], to calculate probabilities of pathogenicity or benignity based on function. Both studies assigned variants to seven functional strength categories (Pathogenic_Strong, Pathogenic_Moderate, Pathogenic_Supporting, VUS or Uncertain, Benign_Supporting, Benign_Moderate, Benign_Strong) consistent with a ClinGen-specified interpretation of ACMG/AMP classification criteria[10]. The functional results from Huang et al.[8] yielded ≥93% sensitivity and ≥95% specificity for established pathogenic and benign missense variants reported in ClinVar[8], whereas the Sahu et al.[9] yielded sensitivity of 89% and specificity of 93%[9]. The functional data from each study were subsequently combined with other sources of genetic and clinical evidence according to ACMG/AMP classification guidelines to classify 6260 (91%) variants in the Huang et al. study[8] and 5819 (89%) variants in the Sahu et al. study[9] as Pathogenic (P), Likely Pathogenic (LP), Likely Benign (LB) or Benign (B).

In this work, we sought to combine the data from the two functional studies to improve the accuracy of the functional measures and to use the combined functional data for improved classification of *BRCA2* VUS. Four models that combined the functional data from the two studies were developed: A "Concordance model" that evaluates consistency of functional results for variants; An "Integrated VarCall model" that merges the raw functional results from the two studies and redefines probabilities of pathogenicity for variants by VarCall[8]; An "Integrated GMM model" that merges raw data and estimates probabilities of pathogenicity using a GMM model; A "Secondary concordance model" that evaluates consistency for variants from the "Integrated VarCall model" and the "Integrated GMM model" (online methods). Results were compared to those from the two original studies[8,9] using established pathogenic and benign clinical standards from ClinVar and functional standards from a well-calibrated homology directed DNA repair (HDR) assay. The "Integrated VarCall model" outperforms the "Concordance model", the individual SGE studies, and other integrated models with 98.9% sensitivity and 100% specificity relative to known standards. Incorporating the improved "Integrated VarCall model" functional interpretation into a ClinGen/ACMG/AMP classification framework of *BRCA2* variants based on ClinGen *BRCA1/2* VCEP specifications classifies 92.8% of all variants and 90.3% of missense variants as P/LP or B/LB.

## Results
### Combined analysis of two *BRCA2* SGE studies
A total of 6383 SNVs were functionally evaluated in both of the original *BRCA2* SGE Huang et al.[8] and Sahu et al.[9] studies and were assigned to seven function-based categories based on probabilities of

pathogenicity or benignity that are consistent with the ClinGen *BRCA1/2* VCEP PS3/BS3 variant classification specifications[8,9]. Importantly, 1041 of these 6383 (16.3%) SNVs yielded discordant functional results in the two original studies. According to current *BRCA1/2* VCEP specifications these discordant variants cannot be assigned functional weight for variant classification. Thus, only 5110 (80%) variants with consistent functional results in the original studies[8,9] were evaluated in the "Concordance model" (Supplementary Data 1, and Supplementary Table 1). In contrast, the "Integrated VarCall model" and the "Integrated GMM model" assigned all 6383 commonly evaluated variants to seven function-based categories (online Methods, Supplementary Data 1). The "Integrated VarCall model" yielded 95.3% and 86.4% concordance by functional category with the Huang et al.[8] and Sahu et al.[9] studies, respectively, with only 303 variants from Huang et al. and 841 from Sahu et al. changing category. In contrast the "Integrated GMM model" yielded 87.7% and 86.5% concordance with Huang et al. and Sahu et al., respectively (Supplementary Table 2, and Supplementary Fig. 1). Of the 6383 commonly evaluated variants, only 5744 (90%) yielded similar results in the two integrated assays and were evaluated in the "Secondary concordance model" (Supplementary Data 1, Supplementary Table 1).

When all P or B category variants (strong, moderate and supporting) from each of the six functional assessment models were compared with ClinVar and HDR control sets, all models yielded close to 100% sensitivity and specificity. To better distinguish between these models, a more conservative approach that included only variants in Pathogenic_Strong and Benign_Strong/Moderate functional categories, with a higher probability of pathogenic or benign activity, was used to evaluate the performance of each model. When compared to 158 P/LP and B/LB missense variant standards in ClinVar that were not influenced by SGE screening results and were commonly evaluated across all six models, the "Integrated VarCall model" showed 98.9% sensitivity and 100% specificity for Pathogenic_Strong and Benign_Strong/Moderate categories, with a single discrepant variant. In contrast, the simple "Concordance model" yielded 95.6% sensitivity and 97% specificity with six incorrectly assigned variants. Likewise, the "Integrated GMM model" and the "Secondary concordance model" yielded 94.5% sensitivity and 98.5% specificity. The original Huang et al.[8] and Sahu et al.[9] studies SGE studies demonstrated 96.7% and 98.9% sensitivity and 100% and 97.1% specificity, respectively (Fig. 1a, and Supplementary Table 3). The better performance of the "Integrated VarCall model" was also observed using missense standards derived from a homology directed DNA repair (HDR) assay. This model yielded 97% sensitivity and 100% specificity, whereas the "Concordance model" gave 87% sensitivity and 98.1% specificity and both the "Integrated GMM model" and "Secondary concordance model" gave 93% sensitivity and 97.7% specificity (Fig. 1b, and Supplementary Table 3). When evaluating the six models with each individual control set, the "Integrated VarCall model" consistently yielded the best performance (Fig. 1c, d, and Supplementary Table 4), while also evaluating 100% of the 6383 common variants (Supplementary Tables 3 and 4).

### Incorporation of the "Integrated VarCall model" functional data into the *BRCA1/2* ClinGen variant classification specifications
Next the impact of the functional model on the complete *BRCA2* ClinGen/ACMG/AMP variant classification process was assessed. The "Integrated VarCall model" functional results (PS3/BS3) were combined with evidence from inactive variants (PVS1), same amino acid alteration (PS1), variant frequency (BA1/BS1), variant rarity (PM2_Supporting), Fanconi Anemia (FA) co-occurrence (PM3/BS2), sequence-based in silico/splicing prediction (PP3/BP4/BP7) and multifactorial likelihood ratio (LR) and/or case-control LR (PP4/BP5) according to the *BRCA1/2* VCEP approved ClinGen/ACMG/AMP classification

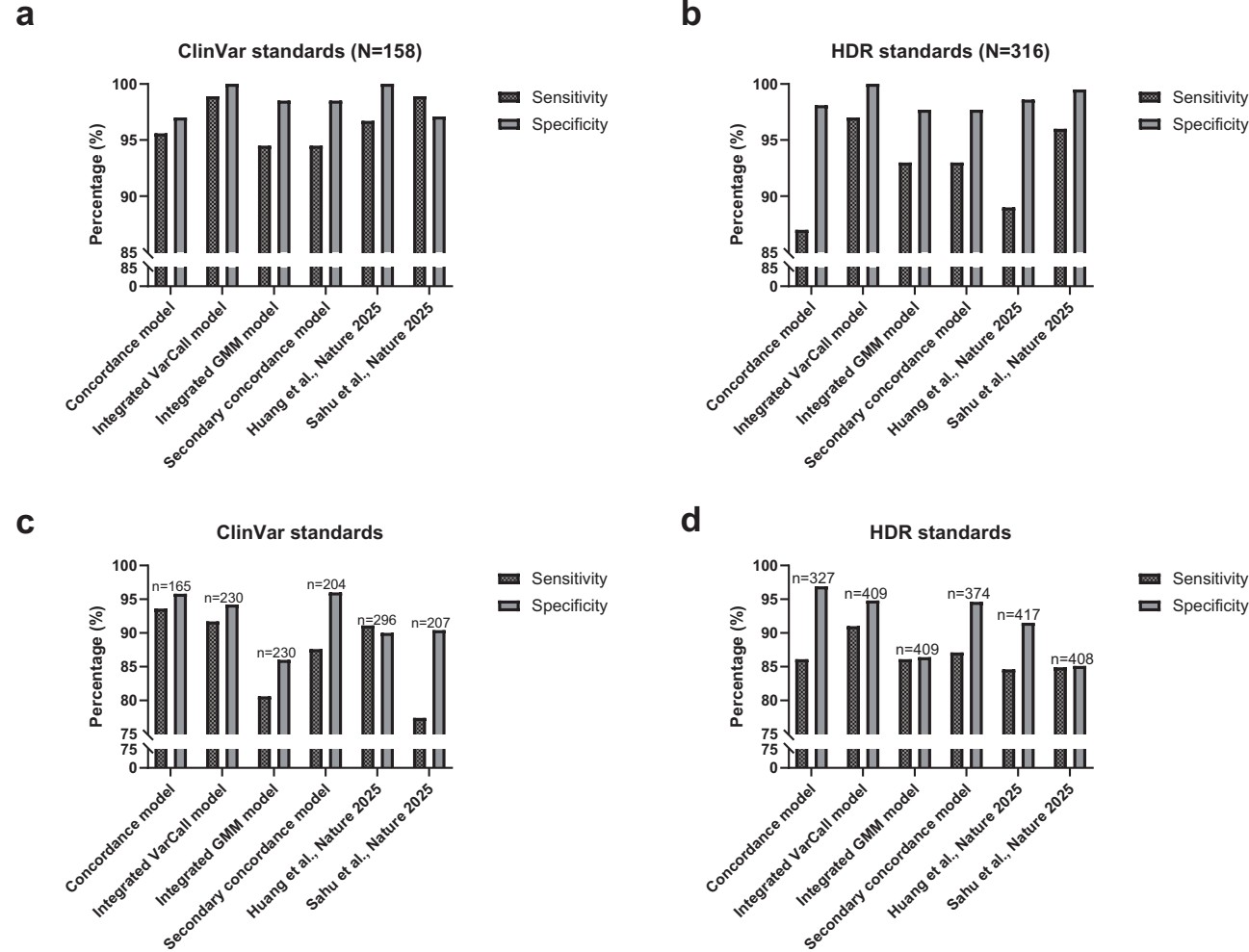

**Fig. 1 | Comparison of the six functional data models with ClinVar and HDR standards. a** Bar chart indicating the sensitivity and specificity of each model compared to ClinVar standards that are common across the six models (N = 158). **b** Bar chart indicating the sensitivity and specificity of each model compared to HDR missense standards that are common across the six models (N = 316). **c** Bar chart indicating the sensitivity and specificity of each model compared to ClinVar standards. The number of ClinVar standards evaluated for each model is displayed above the corresponding bars. **d** Bar chart indicating the sensitivity and specificity of each model compared to HDR standards. The number of HDR standards evaluated for each model is displayed above the corresponding bars. Sensitivity is based on the P_strong function compared to ClinVar P/LP standards and the HDR abnormal category; specificity is based on B_strong/moderate function compared to ClinVar B/LB standards and the HDR normal category. HDR: homology directed DNA repair. Source data are provided in Supplementary Tables 3 and 4.

specifications. On this basis 735 variants, including 237 missense, were classified as P/LP and 5191 variants, including 3769 missense, were classified as B/LB. Thus, 92.8% (5926 of 6383) overall and 90.3% (4006 of 4438) of all missense variants were classified as P/LP or B/LB whereas only 12.2% (780 of 6383) of variants overall and 2.8% (126 of 4438) of missense variants were classified without incorporation of the functional results (Fig. 2, Supplementary Data 2, and Supplementary Table 5). For the 1041 variants with conflicting results from the two original studies, 805 (77.3%) were classified as P/LP (n = 92) or B/LB (n = 713) by the "Integrated VarCall model" (Supplementary Data 2). Importantly, ClinVar or HDR standards with conflicting functional results in the "Integrated VarCall model" were labeled as VUS (discordant) (n = 45) in the final ClinVar/ACMG classification (Fig. 2, Supplementary Data 2, and Supplementary Table 5).

### Breast and ovarian cancer risks associated with functionally characterized missense variants

Missense variants constitute the majority of *BRCA2* VUS in ClinVar. To assess whether the missense variants assigned to the Pathogenic category of the "Integrated VarCall model" conferred increased risks of

breast and ovarian cancer, similar to known pathogenic variants, a series of case-control association studies of pooled variants were performed. Pathogenic_Strong missense variants were associated with moderate to high risks of breast cancer (odds ratio (OR) = 5.46; 95% CI:4.07-7.48) using breast cancer cases that received clinical genetic testing from Ambry Genetics and the AllofUs non-cancer female population as controls adjusting for age and ancestry (Fig. 3a, and Supplementary Table 6). This association was strengthened when restricting to missense variants with posterior probability (PP) > 0.95 (OR = 6.46; 95%CI:4.68-9.15). The other Pathogenic category variants could not be evaluated due to limited numbers of events in either cases or controls (Supplementary Table 6). Missense variants classified as P/LP by the ClinGen/ACMG/AMP guidelines yielded similar results (OR = 7.66; 95%CI: 5.26-11.59), consistent with established pathogenic missense standards or DBD nonsense variants. Moderate to strong associations were also observed using gnomAD v4.1 controls (OR = 5.8; 95%CI: 4.36-7.92) (Supplementary Table 6) and in the combined population-based CARRIERS and BRIDGES studies[1,3] (OR = 3.39; 95% CI:2.20-5.28) (Fig. 3a, and Supplementary Table 6). Likewise, elevated risks of ovarian cancer were observed for Pathogenic_Strong (OR =

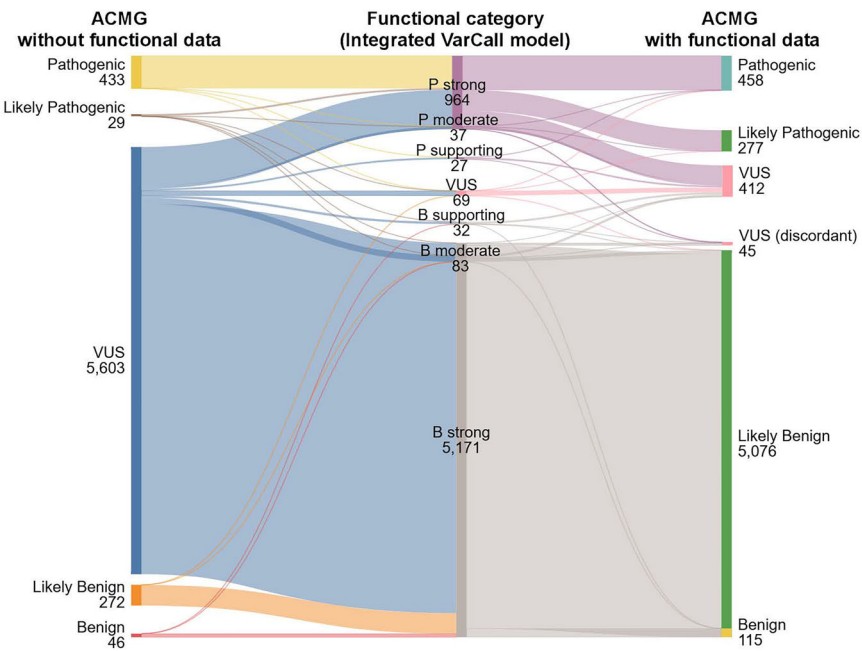

**Fig. 2 | Clinical classification of variants based on the Integrated VarCall model.** Sankey plot illustrating the clinical classification of BRCA2 DNA Binding Domain variants from a ClinGen/ACMG/AMP *BRCA1/2* classification framework with or without Integrated VarCall model functional data. Numbers of single nucleotide variants in functional categories from the Integrated VarCall model are shown in the middle column. ACMG classification without functional data are shown in the left column. Numbers of variants in ACMG classification categories that incorporate the functional data based on the Integrated VarCall model are shown in the right column. Variants with functional results conflicting with ClinVar standards or HDR data were designated as VUS (discordant). ACMG American College of Medical Genetics; VUS variants of uncertain significance; P Pathogenic; B Benign; HDR homology directed DNA repair. Source data are provided in Supplementary Data 2 and Supplementary Table 5.

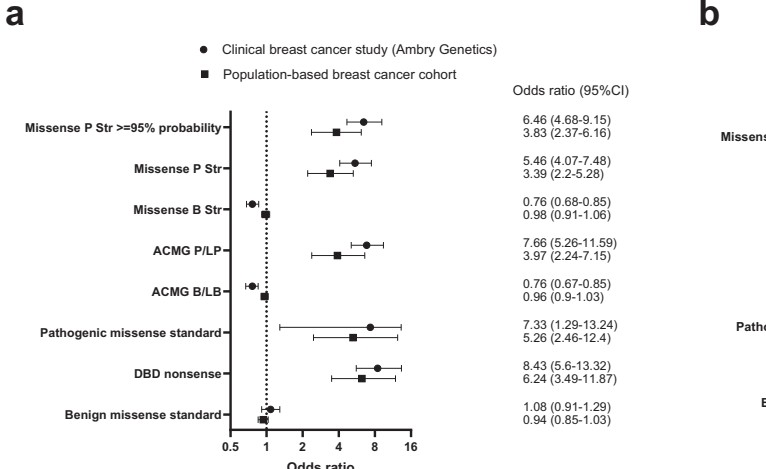

**Fig. 3 | Associations between *BRCA2* missense variants categorized by the Integrated VarCall model and breast and ovarian cancer. a** Log2 (*x*-axis) plot of odds ratios (OR) estimated by comparing frequencies of variants by category in breast cancer cases (N = 376,778) from Ambry Genetics and AllofUs non-cancer female controls (N = 194,610) (black dot) or population-based breast cancer cases (N = 81,073) and controls (N = 83,247) (black square). **b** Log2 (*x*-axis) plot of ORs from ovarian cancer cases (N = 40,762) from Ambry Genetics compared to AllofUs non-cancer female controls (N = 194,610) (black dot). ORs were calculated by logistic regression adjusted for age and ancestry and presented as point estimates with 95% confidence interval. Population-based cases and controls were from CARRIERS[1] and BRIDGES[3]; Pathogenic and benign missense standards were designated by the BRCA1/2 VCEP; Protein-truncating variants were from the BRCA2 DBD; OR odds ratio; CI confidence interval; P Pathogenic; Str strong; DBD DNA binding domain. Source data are provided in Supplementary Table 6.

6.83; 95%CI:4.51-10.40) and ACMG P/LP classified variants (OR = 8.11; 95%CI:4.80-13.88) in the Ambry Genetics/AllofUs analysis (Fig. 3b, and Supplementary Table 6). These results suggest that *BRCA2* missense variants with high PP from the "Integrated VarCall model" are associated with increased clinically relevant risks of breast and ovarian cancer.

## Discussion

In this study, data from two large SGE studies of SNVs in the *BRCA2* DBD were combined in four functional models to improve the accuracy of *BRCA2* variant classification. An "Integrated VarCall model" that combined raw functional scores from the two original studies substantially outperformed a simple "Concordance model", the individual

SGE studies, and other integrated models for improved functional interpretation and ClinGen/ACMG/AMP classification of *BRCA2* variants. The model evaluated all 6383 variants, had 98.9% sensitivity and 100% specificity relative to known standards, and classified 92.8% of all variants and 90.3% of missense variants as P/LP or B/LB using ClinGen *BRCA1/2* VCEP specifications and ACMG/AMP *BRCA2* classification guidelines. Importantly, 77.3% of the 1041 discordant variants from the original two *BRCA2* SGE studies were successfully classified by the "Integrated VarCall model" as P/LP ($n = 92$) or B/LB ($n = 713$). The resulting more accurate classifications of *BRCA2* variants are publicly available at (https://appshare.cancer.gov/BRCA2_Integrated_SGE/).

A substantial number of functionally Pathogenic_Strong variants (242 of 964 (25.1%)) could not be classified due to a lack of information from other sources of evidence. However, because of the strength of the functional evidence, many of these variants will likely be classified once further evidence becomes available. Furthermore, while the two MAVE publications incorporated in this study represent a large proportion of published functional assay results in *BRCA2*, previous smaller scale functional studies of *BRCA2* variants have been published could also potentially contribute to variant classification.

The information provided in this study is expected to substantially improve risk assessment and risk management of individuals found to carry P/LP *BRCA2* DBD variants and will also clarify that many VUS that can now be classified as B/LB are of no relevance to disease. However, further development of evidence codes for pathogenicity or benignity based on clinical and genetic features and additional efforts aimed at variant classification and validation of individual variants are still needed. While the "Integrated VarCall model" proved effective for risk assessment, it remains to be determined whether the functional model is also predictive of response to targeted therapy.

## Methods

This study complies with all relevant ethical regulations.

### Study populations

Individuals with breast cancer ($N = 376,778$) and ovarian cancer ($N = 40,762$) receiving cancer genetic testing by Ambry Genetics were selected as cancer cases. Public reference controls were female non-cancer controls from the AllofUs research program ($N = 194,610$)[11], and females from gnomADv4.1 exomes (excluding UK Biobank) and genomes ($N = 181,964$)[12], and CARRIERS and BRIDGES population-based breast cancer cases (N = 81,073) and unaffected controls (N = 83,247) were from cohort-based nested breast cancer case-control studies or case-control studies[1,3]. The analysis of the clinical-testing cohort was deemed exempt from review by the Western Institutional Review Board. The study was approved by Mayo Clinic IRB (15-007593).

### Statistical analysis

Associations between variant classification groups in BRCA2 and the risk of breast cancer or ovarian cancer were performed for Ambry Genetics female cases and AllofUs (noncancer female) controls using logistic regression adjusted for age (age of diagnosis for cases and age of enrollment for controls) and ancestry, or Ambry Genetics female cases and gnomAD (v4.1 females excluding UK Biobank) using weighted logistic regression with control populations weighted for the relative frequency of different races and ethnicities in the cases. Associations in the population-based CARRIERS and BRIDGES breast cancer cases and unaffected female controls were performed using Fisher's exact test. All tests were two-sided.

### Concordance model

Based on the variants evaluated in both Huang et al. [8] and Sahu et al. [9] studies that had concordant functional categories between the two SGE studies. This included P category variants assigned to any of the pathogenic strong, moderate, or supporting categories in both SGEs ($n = 765$), B category variants assigned to any of the benign strong, moderate, or supporting categories in both SGEs ($n = 4336$), or variants assigned to the VUS category in both SGEs ($n = 9$). A total of 5110 concordant variants were evaluated in the Concordance model.

### Integrated VarCall model

VarCall is a class of Bayesian hierarchical models with context-specific measurement models that embed a Gaussian two-component mixture model for the variant effects. The formulation used here is based on analysis of *BRCA2* variants by Huang et al. [8]. Variants were each assigned a binary indicator of pathogenicity status, deterministically if assumed known and probabilistically if not. Silent variants were assumed benign, and nonsense variants pathogenic. The measurement model adjusted for batching by including replicate by exon level location and scale random effects, and included t-distributed error terms to allow for outliers. The JAGS language[13] was used to specify and fit the VarCall model using a Markov chain Monte Carlo (MCMC) algorithm. All related computations were carried out in the R programming language[14]. A prior probability of pathogenicity of 0.2 for variants in the DNA binding region was used based on AlphaMissense predictions that 22.7% of missense variants in the BRCA2 DBD are pathogenic[8] (Supplementary Table 7). We also applied prior probability of pathogenicity of 0.1, and the resulting functional categories have 99.5% concordance compared to prior probability of 0.2 (Supplementary Table 8). Using the MCMC output, the Bayes Factor (BF) in favor of pathogenicity for each variant was computed. The thresholds for the Bayes factor based on strength of evidence of pathogenicity or benignity (P_Strong/Moderate/Supporting, VUS, B_Strong/Moderate/Supporting) were derived from the Bayesian interpretation of the ACMG/AMP guidelines[15]. Because the HAP1[8] and ES cell[9] studies displayed similar functional score ranges and similar sensitivity/specificity relative to ClinVar standards, the ES cell data were normalized and rescaled relative to the HAP1 cell data. The duplicate raw score values for every variant from the ES cell study were combined with the raw scores from the HAP1 cell study (at least three replicates) for each of the 6383 commonly assessed variants.

### Integrated GMM model

The dropout frequency of SNVs was defined as a functional score and incorporated into a two-component Gaussian mixture model (GMM) to estimate the probability of impact on function (PIF) as described in Sahu et al. [9]. The function scores were integrated from two replicates of ES cell-based study and at least three replicates of HAP1 cell-based study. The PIF was calibrated by treating synonymous variants as functional and nonsense variants as non-functional. A prior pathogenicity probability of 0.2 was assigned to variants in the DNA-binding region (Supplementary Table 7), and the odds of pathogenicity were then calculated to classify SNVs into seven categories based on the strength of evidence for pathogenicity or benignity.

### Secondary concordance model

Based on the Integrated VarCall model and the Integrated GMM model, variants that have consistent functional categories between these two models were included in the Secondary concordance model: P category variants (in any of the pathogenic strong, pathogenic moderate, or pathogenic supporting category across the Integrated VarCall model and the Integrated GMM model) (n = 964), B category variants (in any of the benign strong, benign moderate, or benign supporting) (n = 4768), or VUS category variants (n = 12), total 5744 concordant variants were evaluated in the Secondary concordance model.

## ClinGen/ACMG/AMP classification

The ACMG/AMP rule-based framework was based on ClinGen ENIGMA (Evidence-based Network for the Interpretation of Germline Mutant Alleles) BRCA1 and BRCA2 Expert Panel Specifications to the ACMG/AMP Variant Interpretation Guidelines for BRCA2 Version 1.2.0 (https://cspec.genome.network/cspec/ui/svi/doc/GN097), which includes evidence from functional evaluation, computational prediction, population frequency, and other data. Criteria were weighted as either very strong (+8), strong (+4), moderate (+2), or supporting (+1) for evidence for pathogenicity, or stand-alone (−8), strong (−4), moderate (−2), or supporting (−1) for evidence for benignity. Criteria used included PVS1_variable and PVS1_variable[RNA] based on ClinGen VCEP specifications table 4 and supplementary table 2-3; PS1_variable based on literature[16] and Ambry Genetics data), applied per ClinGen VCEP specifications table 5 and appendix table 17; PS3_variable based on the VarCall model capped at +4 points; PM2_Supporting based on absence from gnomAD v4.1 non-UKBB (This is updated from gnomAD v2.1 and v3.1 that were listed as the preferred datasets in the *BRCA1/2* VCEP specifications); PM3_variable based on ClinGen supplementary table 9 and Ambry Genetics data, applied per ClinGen specifications table 6; PM5(PTC)_variable based on ClinGen VCEP specifications table 4; PP3_variable, PP3_Moderate was applied for variants with BayesDel ≥0.3 based on ClinGen supplementary table 12 and 14 and PP3 was applied for variants meeting ClinGen VCEP SpliceAI specifications; PP4_variable based on likelihood ratio derived from multifactorial likelihood clinical data or case-control data[7,17–21], weight applied per ClinGen VCEP specifications; BA1, BS1, and BS1_Supporting per ClinGen VCEP specifications, gnomAD v4.1 filter allele frequency >0.001, >0.0001, and 0.00002 respectively; BS2 based on data from CARRIERS and Ambry Genetics and applied per ClinGen appendix H; BS3_variable based on the VarCall model and is capped at −4; BP4 based on ClinGen VCEP BayesDel and SpliceAI specifications; BP5_variable based on likelihood ratio derived from multifactorial likelihood clinical data or case-control data[7,17–21], weight applied per ClinGen VCEP specifications; BP7 and BP7_Strong[RNA] based on ClinGen VCEP specifications Fig. 1 and supplementary table 2-3. The final ACMG classification includes benign (B) (≤−7 points), likely benign (LB) (between −6 and −2 points), pathogenic (P) (≥+10 points), likely pathogenic (LP) (between +6 and +9 points), and variant of uncertain significance (VUS) (between −1 and 5 points). The 5926 variants that incorporated the "Integrated VarCall model" functional data to be classified either as LP/P or B/LB by the ACMG/AMP framework represented 92.8% of the total 6383 variants that evaluated in current study. The remaining 457 variants were either staying as VUS due to not enough evidence points (n = 412) or have conflicting with ClinVar standards (n = 8) or HDR functional results (n = 37).

## Reporting summary

Further information on research design is available in the Nature Portfolio Reporting Summary linked to this article.

## Data availability

All data presented in this study are publicly available in GEO (GSE270424[GEO Accession viewer], GSE248438[GEO Accession viewer]). Source data are provided in Supplementary tables and Supplementary Data.

## Code availability

All related code for functional score calculations using the VarCall and GMM models was deposited in Github (najiemayo/Couch_SGE_BRCA2_MAVE_Mayo_NCI_joint_analysis)

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

## Acknowledgements

This study was funded by NIH grants R35CA253187, U24 CA258058, a specialized program of research excellence (SPORE) in breast cancer

P50CA116201, and grants from the Breast Cancer Research Foundation (BCRF), the BRCA research & cure alliance and Mayo Clinic Department of Laboratory Medicine and Pathology (F.J.C.). This study was supported in part by the Intramural Research Program of the National Institutes of Health (NIH) (S.K.S.). The contributions of the NIH author(s) were made as part of their official duties as NIH federal employees, are in compliance with agency policy requirements, and are considered Works of the United States Government. However, the findings and conclusions presented in this paper are those of the author(s) and do not necessarily reflect the views of the NIH or the U.S. Department of Health and Human Services.

## Author contributions

C.H., S.S., W.C., M.G., M.E.R., M.F.B., R.K., T.P., J.N., H.H., J.N.W., K.N.N., S.Y., N.J.B., S.M.D., A.N.M., E.S.I., S.K.S., and F.J.C. all contributed to analyzing the data and writing the paper.

## Competing interests

R.K., T.P., and M.E.R. are employees at Ambry Genetics. The remaining authors declare no competing interests.
