## [Transparent Peer Review file · Nature Communications]

Combining multiplexed assays of variant effect for enhanced BRCA2 variant classification

Corresponding Author: Dr Fergus Couch

Version 0:

Reviewer comments:

Reviewer #1

(Remarks to the Author)

The authors have taken the results from two recent BRCA2 MAVE studies. They have improved the concordance of these functional results with ClinVar classifications by re-analyzing the raw data using a series of combined models. The best performing being the Integrated VarCall model.

Results were then combined with a few data sources to generate a proposed classification according to the BRCA1/2 VCEP approved ClinGen model.

The title should be more specific about the topic – the main purpose of the paper was about whether combining MAVES improved results, and that is not obvious from the title as written.

Likewise, in the abstract there is need for more detailed information about what was in the model/s and whether it was combined functional data only that improved the classification, or the other aspects of the model.

Main text

When referring to ClinGen VCEP “guidelines” the appropriate term is “specifications”. Likewise, when referring to the ACMG/AMP, the appropriate term is guidelines (not models).

The comment that the expert panel-based classifications are “heavily dependent on functional evaluation” is not justified. There are many other evidence types specified for use at strong pathogenic or strong benign evidence. Rephrase to state the obvious, that functional assay evidence at strong level can be helpful for any classification. A more appropriate justification for this study is that conflicting functional evidence cannot be applied at any weight, according the current VCEP specifications for BRCA2. If this study helps remove the discordances between the two MAVES then that goes one step towards addressing the issue that currently ~20% of the variants assayed by both MAVES cannot have functional data applied to them when following VCEP specifications since the results do not agree in direction of evidence (this % is based on the data provided in Supplementary Table 1).

The following sentence needs to be rephrased so that the reference to the ES cell study (and number of variants classified) is placed before the classification levels “The data were subsequently combined with other sources of genetic and clinical evidence in a ClinGen VCEP/ACMG/AMP classification model to classify 6260 (91%) variants from the HAP1 study as Pathogenic (P), Likely Pathogenic (LP), Likely Benign (LB) or Benign (B)4 and 5819 (89%) variants from the ES cell study5.”

It would seem that the authors did not actually follow the ClinGen VCEP specifications for BRCA2. E.g. computational prediction evidence for missense variants was applied at moderate evidence, where the specifications state that computational evidence should be applied at supporting evidence (for variants within a functional domain).

There are also discrepancies between the main text and Supplementary Table 5, which contains the information on these classifications:

- Variant rarity is incorrectly called PM1 in the main text, but presented as PM2 at supporting level in ST5
- Sequence based in silico/splicing predictions is incorrectly called PM3 in the text, and is presented as PP3 in ST5.
- As noted above, PP3 is also applied at moderate level for sequence-based predictions for missense variants, which differs from the ClinGen BRCA1/2 VCEP guidance

- Many additional pieces of evidence are included in ST5 are not described in the main text (PVS1, PS1, PP4, BS2, BP5 codes)
- PM5 in the main text is referred to as case-control data. The ClinGen VCEP specifications for BRCA2 state that case-control odds ratio data can be applied under PS4, and that only case-control Likelihood ratio data is preferred and should be under PP4 (or BP5). PM5 is not mentioned in Supplementary Table 5. It is unclear if the case-control data described later in the text was used in the final classification.

Additional methods descriptions are required for what evidence is used and where it came from (publication, internal data etc), including code strengths drawn from the specifications.

Further, to accurately justify the value of functional data for increasing the classification rate, the authors must show the classifications using all other data first, and then the change in classifications by adding the functional data. Figure 2 legend needs to be expanded to be clear that the central pillar highlights the functional weight but that other information was included together with the functional data to reach the final classification shown in the figure. Refer to Supplementary Table 5 in the legend, and this table should be updated to provide more information on the sources of information (as noted above). There appears to be a problem with the bottom of Figure 2 being repeated?

It would also be important to repeat the sensitivity and specificity including functional data at other (lower) strengths. All evidence is useful in variant classification, from supporting to very strong evidence. Lines 92-97 suggest that only Pathogenic_Strong and Benign_Strong/Moderate are useful in classification. In a practical setting, functional results are combined with all other evidence for potential classification - which is what the authors have done in their study, despite saying that only the highest categories have the potential for classification.

Can the authors justify why they used a prior of 0.2. Standard acmg/amp assumes a prior of 0.1 (that was the justification for the bayesian modelling paper), and this prior was carried over for the ClinGen VCEP specifications as well.

It is unclear the spread of classifications for missense variant standards in ClinVar, total 158. Given these data are used to justify the Integrated VarCall model as superior, it is important to know how many were P/LP and how many were B/LB, to ensure sufficient controls on both sides of the equation were included. It should be noted that filtering on missense and ClinVar standards in column K of supplementary Table 5 does not yield 158 variants (the number returned is 125).

The OR calculations show good concordance for VarCall results of Pathogenic Strong with both pathogenic missense standards and DBD protein truncating variants. Although the number of variants with Pathogenic Moderate/Supporting VarCall results is quite small, it would be valuable to include those results also in the OR plots. The study would benefit from providing guidance about how to use the Pathogenic Moderate/Supporting and Benign Supporting categories (apply evidence at these levels, or exclude as evidence).

Last, the website produced to display the data requires more work to be transparent about what is displayed, and that the information supporting the final classification is downloadable (although with a different table number ST3). Further explanation is required in ST5 (and the download file) to explain what the columns mean, and how they were used e.g. spliceR? ; PVS1 point "replace with PS3"?. How were the PS1 points derived? What is the reference variant? Where does the PP4 information come from? Note there are multiple typographical errors in the headers.

Did the authors use PS3 (functional data) from other studies? Unclear

References

- in addition to reference 1 (from this group), it would be pertinent to refer to the other large-scale population-based breast cancer study providing risk estimates for BRCA2 (and indeed a study used to generate odds ratio data).
- Check references are appropriate, eg the introduction of the ClinGen BRCA1/2 VCEP (ref 3) appears to refer to a 2019 paper about a multifactorial likelihood model

(Remarks on code availability)

Reviewer #2

(Remarks to the Author)

Hu and Sahu et al. present several methods to integrate the raw data from two saturation genome editing MAVEs on the gene BRCA2. These methods are benchmarked against established standards from ClinVar and an independent HDR assay. The authors observe that an Integrated VarCall method is the best performer and also generates a score for all variants in common between the two input MAVEs. ACMG/AMP variant classification is performed to ask how the Integrated VarCall method would alter final pathogenicity determination of each variant. As a further validation, the authors calculate Odds Ratios to ask whether variant classification by the Integrated VarCall method identifies variants which impact cancer risk and find that benign variants do not confer risk while pathogenic variants do confer risk. Overall, this is a study that will likely be of interest to a broad audience, particularly cancer biologists, geneticists, and clinical geneticists. There are a number of clarifications that, in my opinion, would be necessary to make prior to publication. Please see below.

Major concerns

-It was a bit surprising how little information was given on the individual MAVE scoresets other than the single paragraph starting on line 60. A major question I had all of the way through the manuscript is how the integrated models perform relative to the individual input MAVE scoresets. It appears, based on the limited information given, that both sensitivity (89%/93%  98.9% ?) and specificity (~95%/93%  100% ?) are improved, but it is unclear if we can say that for certain. For the integrated scoresets, it is clarified that only Pathogenic_Strong and Benign_Strong/Moderate were used in calculating metrics. Was this the case for the individual MAVE scoresets? I think to alleviate this issue and make it clear to readers that the integrated VarCall method is superior, the individual MAVE scoresets should be included into at least Fig 1. A direct comparison of sensitivity and specificity between the individual MAVE scoresets and the integrated models would make it clear how much of an improvement in performance is actually obtained. The authors might further consider supplemental Sankey plots for the individual MAVE scoresets to directly compare with the Integrated VarCall classifications. It might also be interesting to describe in some way, perhaps a supplemental table or Venn diagram, how many and which variants are classified differently by the Integrated VarCall scoreset relative to each input MAVE scoreset. In addition to the average reader of this manuscript, it is not hard to imagine that the appropriate Clingen VCEP would want this information in order to decide whether to adapt their rulesets to using the Integrated VarCall for assigning PS3/BS3.

-It is not clear to me why only Pathogenic_Strong and Benign_Strong/Moderate variants were used in calculating specificity and sensitivity for each model. I would kindly request the authors to justify this. Unless there is a specific reasoning that was not clear to me from reading this manuscript, it would seem most fair to include all variants scored by each model within each truth set.

Minor concerns

-There are no accession number(s) given for GEO in the Data Availability...is the data available?

-In the Fig 2 Sankey plot, it appears that around 20% of the variants that receive an Integrated VarCall score of PS3 strong end up with a final ACMG/AMG classification of VUS. It would be helpful if the authors could comment on this as it seems counterintuitive.

-There seems to be a weird extra segment in the Sankey plot at the bottom of the figure. A portion of the silent variant node appears to have been duplicated? This should be corrected.

(Remarks on code availability)

Reviewer #3

(Remarks to the Author)

(Remarks on code availability)

Reviewer #4

(Remarks to the Author)

As the authors outline in their introduction, accurate variant interpretation in cancer susceptibility genes can have a large impact on patient management. Functional assay data are particularly useful for genes such as BRCA2 where there is generally a lack of phenotypic data to support variant interpretation. Functional data to assist with accurate variant interpretation is of interest and utility to the cancer susceptibility genetics community.

This work draws on previously published functional assay results and aims to optimise the interpretation of these results through the combination of results from multiple assays. In general, this is a thoughtful and helpful exploration of multiple methods for interpreting this data. My main concerns are how the data generated through the methods described can be incorporated into existing clinical variant interpretation frameworks with appropriate weighting and avoiding circularity. Please find specific queries and suggestions below:

Intro

- Suggest to add comment about importance of reclassification of VUS to benign/likely benign (as well as to pathogenic)
- Please revise final sentence "are listed on the 58 ClinVar website (April 2, 2025) along with proposed classification of variants by genetic testing 59 laboratories and over 5000 BRCA2 VUS and variants with conflicting interpretations." – doesn't make sense to me, implies that VUS is not a proposed classification
- Additional references needed where VCEP and ACMG/AMP guidance mentioned

Methods

- Unclear why variants with conflicting results were excluded from the concordance model, does the rate of this not provide

information? What counts as conflicting?

- Please explain why only 90% of variants reviewed for the secondary concordance model if all variants looked at for integrated VarCall and GMM model?
- Why was pathogenic_moderate evidence not incorporated into assessment? Could lead to classification if other evidence available (supporting could also, albeit less likely). I can see that there were fewer variants falling into these categories so was there concern that doing a “banded” analysis of different strength classifications might be underpowered?
- Were the pathogenic and benign control variant classifications reached without functional data? If not, the potential limitation of circularity should be mentioned, particularly if involve assays similar to those studied
- On a similar note, although concordance of the model with an alternative (HDR) is notable, it should be made clear that this is not equivalent to clinical classifications incorporating non-functional data
- Where PM1, PM3 and PM5 codes cited should be PM2, PP3 and PS4 respectively
- Need to explicitly describe whether overlapping data used between VarCall and the additional data used to produce “ACMG” classifications
- It would be helpful to explain the VarCal pathogenicity categories earlier, including more detail about how they were calculated in the supplementary methods and how the thresholds were derived, since I can't see how they fit within in the ACMG Bayesian model which is often used to quantify other types of evidence
- Only a few of the ACMG/AMP codes are mentioned in the online methods – is this deliberate? Please explain how classification was done and by who – i.e. what data was available to the variant classifiers. This last paragraph of online methods also needs rereading for grammar
- Please explain justification for using a prior probability of 0.2 for the integrated models

Results

- Is there a supplementary table 5?
- “Subjected to genetic testing” I find a curious phrase to describe clinic testing in the legends of the supplementary tables – consider updating to “in whom genetic testing was undertaken”
- Quite different results for odds ratios using different control datasets – how confident are you that the regression has adjusted sufficiently for ethnicity mismatching between cases and controls? I can't see why the OR would be lower in All of US, if anything if confirmed no cancer phenotype should be higher? If Ambry contained cases with a mixture of ethnicities, unclear why only Europeans from All of Us were selected as controls.
- Age matching mentioned in online methods, please present data

General

- This may be due to the formatting required by the journal but the main text would be more readable if put under sub headings (online methods has clearer structure)

(Remarks on code availability)

Version 1:

Reviewer comments:

Reviewer #1

(Remarks to the Author)

This review was performed in conjunction with an Early Career Researcher, and represents a combined report.

The authors have addressed many concerns and issues raised from the initial review.

Please check sentences are correct after track changes are included, eg

From Abstract: Varcall is mentioned twice in the same sentence, which appears redundant? “An “Integrated VarCall Model” that combined raw functional data from 6383 variants evaluated in both SGE studies in a VarCall probability-based model yielded 98.8% accuracy and out-performed the original SGE studies and three other combined data models”

From Discussion: An “Integrated VarCall model” that combined raw functional scores from the two studies of substantially outperformed a simple “Concordance model”, the individual SGE studies, and other integrated models for improved functional interpretation and ClinGen/ACMG/AMP classification of BRCA2 variants.

Additional comments in response to the authors are provided, each below the original comment and author response.

Original Comment. It would seem that the authors did not actually follow the ClinGen VCEP specifications for BRCA2. E.g. computational prediction evidence for missense variants was applied at moderate evidence, where the specifications state that computational evidence should be applied at supporting evidence (for variants within a functional domain).

Response: We applied the ClinGen BRCA1/2 VCEP specifications as listed in “SupplementaryTables_V1.2_2024-11-18.xlsx “ in the ClinGen system. Regarding bioinformatics tools we applied the rules as described in Supplementary Table 12 (ST12) “LR towards pathogenicity as predicted by different bioinformatic missense tools, using functional assay datasets as reference” and Supplementary Table 14 (ST14) “LR towards pathogenicity as predicted by BayesDel across three categories, using an updated functional assay dataset as reference”. Since both ST12 and ST14 set “Moderate evidence of

pathogenicity" (Mod P) for BayesDel ≥ 0.3 , following recalibration of the BayesDel thresholds for BRCA1/2, we applied moderate evidence strength for BayesDel PP3 (≥ 0.3). However, ST12 applies "Supporting evidence of benign" (Supp B) for BayesDel < 0.3 and ST14 applies "moderate evidence of benign" (Mod B) for BayesDel ≤ 0.18 . Here, we took a conservative approach and applied "no evidence for BayesDel 0.18-0.3, and applied Supp B for BayesDel ≤ 0.18 . We also applied all other ClinGen VCEP specifications as outlined in ClinGen except that we have now used gnomAD v.4.1 for estimation of population frequency under the BA1 and BS1 specifications because the BRCA1/2 VCEP has recently voted to adopt this strategy.

Additional comments: While the ST12 and ST14 show LRs that fall into the Moderate evidence of pathogenicity, there is only reference to PP3 code being applied in all other places in the specifications. On the ClinGen Specifications website there is no allowance described for modification of PP3 to other evidence categories. Additionally, the recent publication by the VCEP (PMID: 39142283) states in the PP3 section: "The VCEP opted, conservatively, to apply this evidence type at supporting weight only (Table 2)."

Filtering on ST6, this will only change classification for 24 variants that are LP with 6 points, including 2 points from PP3. It is particularly important to use the conservative application of PP3/BS4 for this paper specifically, since the bioinformatic tools were calibrated against functional data results, and there will be circularity due to correlation of the MAVE results with the original calibration reference set. [Some might argue, validly, that PP3/BP4 should not be used at all]

Please note the change to using gnomAD v 4.1 specifically in the methods section for ClinGen/ACMG/AMP classification, to avoid confusion for readers who refer to the specifications version as stated in the paper

Original comment: Additional methods descriptions are required for what evidence is used and where it came from (publication, internal data etc), including code strengths drawn from the specifications.

Response: We have thoroughly revised the method section "ClinGen/ACMG/AMP classification" (page 8-9) to describe in detail the evidence used, the respective sources, and the code strengths derived from the specifications. We also added extra columns and detailed footnotes in Supplementary Table 6 to describe the strength of evidence applied and the sources.

Additional comments: This section is greatly improved. There are 2x PS3 statements in a row. Does this statement refer to PS1 (Splicing) rather than PS3? "PS3), applied per ClinGen VCEP specifications table 5 and appendix table 17". For Supplementary Table 6, please add a marker to the footnote for ACMG/AMP criteria (as is done for HDR and ClinVar standards. Potentially separate ones for each code. E.g. it was not clear that the reference numbers for LR sources were contained within description for PP4/BP5. There is no indication of the sources for application of RNA codes in the Supplementary Table or the manuscript text. There is a typographical error in header for column AT, and the header needs more explanation – what does before " _ " refer to?

Original comment: Further, to accurately justify the value of functional data for increasing the classification rate, the authors must show the classifications using all other data first, and then the change in classifications by adding the functional data. Figure 2 legend needs to be expanded to be clear that the central pillar highlights the functional weight but that other information was included together with the functional data to reach the final classification shown in the figure. Refer to Supplementary Table 5 in the legend, and this table should be updated to provide more information on the sources of information (as noted above). There appears to be a problem with the bottom of Figure 2 being repeated?

Response: We agree with the reviewer's point that showing the impact of the added functional data on the classification rate will improve the justification of the value of the functional data. We have now included results with and without the functional data in additional columns "Total ACMG points without PS3/BS3" and "ACMG classification without PS3/BS3" in Supplementary Table 6. We have also added a sentence in the text (page 4, last paragraph) to reflect the increased classification rate "Thus, 91.6% (5849 of 6383) overall and 88.5% (3926 of 4438) of all missense variants were classified as P/LP or B/LB whereas only 12.2% (781 of 6383) of variants overall and 2.9% (127 of 4438) of missense variants were classified without incorporation of the functional results". We modified Figure 2 by adding a new pillar showing the ACMG classification without incorporation of functional data and by updating the legend for Figure 2 as requested to show that the central pillar reflects only the functional weight. We also corrected the error at the bottom of Figure 2. As noted above, the ACMG classification framework (Supplementary Table 6) has also been updated.

Additional comments: The addition of pillar 1 showing initial classification highlights the difference adding in functional data can make. The impact is less obvious by going through the variant type pillar. It looks like the figure is now addressing two different points, 1) how likely variants of a certain type are to get certain functional assay outcomes and their subsequent classification, 2) how classification changes with the addition of these functional data from this study. If the main purpose of the figure is to show 2), reclassification impact of these functional data, consider removing variant type (which is available in ST6).

Original comment: Did the authors used PS3 (functional data) from other studies? Unclear

Response: We did not use any other functional studies for applying PS3. In this manuscript PS3 is based solely on the "Integrated VarCall model" functional data. However, if the "Integrated VarCall model" results are inconsistent with HDR or

ClinVar standards, then we labeled the variants as “VUS (discordant)” to be conservative regarding the ACMG classification. (page 5, 1st paragraph).

Additional comments: Suggest adding a comment on this in the discussion as a limitation of the study. While the two MAVE publications incorporated in this study represent a large proportion of published functional assay results in BRCA2, previous work has been published in this area and may contribute to variant classification as well (i.e. not all changed classifications are new when incorporating other functional assay work).

Original comment: the website produced to display the data requires more work to be transparent about what is displayed, and that the information supporting the final classification is downloadable (although with a different table number ST3).

Response: The website (https://brca2-variants-explorer.shinyapps.io/integrated_model/) has been updated to make it easier for those accessing the site to view and/or obtain information. In particular, the data are now downloadable through a specific button link. That table has detailed explanations of the various evidence codes and functional scores for each variant. This corresponds to Supplementary Table 6 in the manuscript. In addition, the website provides a visual representation of the ClinGen/ACMG/AMP classification and the integrated model functional scores for each variant. We believe that viewers will find this NIH supported website very useful.

The website is improved, but requires more editing to distinguish between what is a functional assay “call” and strength of evidence versus final classification using ACMG/AMP guidelines.

It is not clear if the colored boxes refer to final classification. This is important since VUS is also used as a category for the functional code assignment.

e.g. “Variants were classified based on the functional scores obtained from the Integrated VarCall Model” should rather read something like “were assigned an ACMG/AMP functional code and strength based on functional scores from the VarCall Model”

e.g. For ACMG/AMP classification, the functional evidence was combined with other available evidence, based on ClinGen ENIGMA BRCA1 and BRCA2 VCEP Specifications.

- and please provide the specifications version.

For the data field description, clarify that the ACMG/AMP classification was following ClinGen VCEP specifications for this gene (and version number).

The column header Integrated Model Classification should be changed to something like “Functional code assigned based on Integrated Model” or “Integrated Model Category”.

(Remarks on code availability)

Review of code is outside the expertise of the reviewers.

Reviewer #2

(Remarks to the Author)

The authors addressed all of my concerns in their revised manuscript.

(Remarks on code availability)

Reviewer #3

(Remarks to the Author)

(Remarks on code availability)

Reviewer #4

(Remarks to the Author)

The authors have satisfactorily addressed my comments and the manuscript is much clearer.

They have explained why a prior of 0.2 was used for the DNA binding domain in their rebuttal, but I would recommend they include this rationale in the main text of the manuscript.

(Remarks on code availability)

Version 2:

Reviewer comments:

Reviewer #1

(Remarks to the Author)

I co-reviewed this manuscript with one of the reviewers who provided the listed reports. This represents a combined report.

The authors have addressed many concerns and issues raised from the previous review. The only adjustment that needs attention is the applied supporting weight of evidence for PP3/BP4.

Author Response: We have now applied the conservative “supporting” weight of evidence for PP3/BP4. All related text, methods, tables, and figures now reflect this change.

Reviewer comment: In the supplemental table ST6 (632986_2_data_set_12263831_t90cvc.xlsx), 2 points are still applied for PP3 based on BayesDel predictions. This is carried into the sum of points and may impact the overall classification numbers. Filtering in ST6 matches the current numbers in the manuscript, which means that this change hasn't been taken through all of the data, although statements are updated in the text.

(Remarks on code availability)

Reviewer #3

(Remarks to the Author)

(Remarks on code availability)

RESPONSE TO THE REVIEWER COMMENTS

We thank the reviewers for their constructive comments and valuable suggestions. These have helped us to significantly improve the quality of the manuscript. We have addressed all the concerns of the reviewers and have made significant changes in the manuscript as described below in our point-by-point response to the comments.

Reviewer #1:

The authors have taken the results from two recent BRCA2 MAVE studies. They have improved the concordance of these functional results with ClinVar classifications by reanalyzing the raw data using a series of combined models. The best performing being the Integrated VarCall model.

Results were then combined with a few data sources to generate a proposed classification according to the BRCA1/2 VCEP approved ClinGen model.

The title should be more specific about the topic – the main purpose of the paper was about whether combining MAVES improved results, and that is not obvious from the title as written.

Response: We thank the reviewer for pointing this out. The title has been revised to “Combining multiplexed assays of variant effect for enhanced *BRCA2* variant classification”, which we feel more appropriately represents the objective and scope of the paper.

Likewise, in the abstract there is need for more detailed information about what was in the model/s and whether it was combined functional data only that improved the classification, or the other aspects of the model.

Response: The manuscript describes the reevaluation of results from two BRCA2 MAVE studies with the intent of improving the final functional data so that we can achieve better ClinGen/ACMG/AMP classification of BRCA2 variants. We agree that the original abstract talked about improved classification even though we were mainly talking about improved functional data. We have modified the abstract to now make clear that we are comparing and integrating functional data and reporting on improved functional performance relative to standards. At the end we note that we combined all of the functional model results, with common data from other sources of evidence, using ClinGen/ACMG/AMP-based classification guidelines, to show how improved functional data improves variant classification.

Main text

When referring to ClinGen VCEP “guidelines” the appropriate term is “specifications”. Likewise, when referring to the ACMG/AMP, the appropriate term is guidelines (not models).

Response: We thank the reviewer for pointing this out, and we have corrected the respective terms to reflect the appropriate usage of ClinGen VCEP specifications and ACMG/AMP guidelines throughout the manuscript.

The comment that the expert panel-based classifications are “heavily dependent on functional evaluation” is not justified. There are many other evidence types specified for use at strong pathogenic or strong benign evidence. Rephrase to state the obvious, that functional assay evidence at strong level can be helpful for any classification.

Response: We thank the reviewer for pointing this out. We have removed this comment from the Introduction.

A more appropriate justification for this study is that conflicting functional evidence cannot be applied at any weight, according to the current VCEP specifications for BRCA2. If this study helps remove the discordances between the two MAVES then that goes one step towards addressing the issue that currently ~20% of the variants assayed by both MAVES cannot have functional data applied to them when following VCEP specifications since the results do not agree in direction of evidence (this % is based on the data provided in Supplementary Table 1).

Response: We thank the reviewer for this insightful comment. We have now mentioned in the Results Section (page 3, last paragraph) that 1041 of 6383 (16.3%) of variants analyzed by both of the original MAVE functional studies showed discrepancies when comparing the two studies. We comment, as noted by the reviewer, that these variants should be excluded from having functional weights applied for variant classification according to the BRCA1/2 VCEP specifications when looking at the original studies alone or in the “Concordance model”. We note that the proposed “Concordance model”, which excludes these discrepant variants, refines the list of variants that can be used for variant classification, whereas the two “Integrated” models that combined the raw data from the two original MAVES for a single readout allows retention of these discrepant variants.

The following sentence needs to be rephrased so that the reference to the ES cell study (and number of variants classified) is placed before the classification levels “The data were subsequently combined with other sources of genetic and clinical evidence in a ClinGen VCEP/ACMG/AMP classification model to classify 6260 (91%) variants from the HAP1 study as Pathogenic (P), Likely Pathogenic (LP), Likely Benign (LB) or Benign (B)4 and 5819 (89%) variants from the ES cell study5.”

Response: We have rephrased the sentence to be clear that we are referring to two separate models, one from each of the original MAVE studies. The sentence reads as follows “The functional data from each study were subsequently combined with other sources of genetic and clinical evidence according to ACMG/AMP classification guidelines to classify 6260 (91%) variants in the Huang et al., 2025 study⁸ and 5819 (89%) variants in the Sahu et al., 2025 study⁹ as Pathogenic (P), Likely Pathogenic (LP), Likely Benign (LB) or Benign (B)” (Page 3, 1st paragraph).

It would seem that the authors did not actually follow the ClinGen VCEP specifications for BRCA2. E.g. computational prediction evidence for missense variants was applied at moderate evidence, where the specifications state that computational evidence should be applied at supporting evidence (for variants within a functional domain).

Response: We applied the ClinGen BRCA1/2 VCEP specifications as listed in “SupplementaryTables_V1.2_2024-11-18.xlsx” in the ClinGen system. Regarding bioinformatics tools we applied the rules as described in Supplementary Table 12 (ST12) “LR towards pathogenicity as predicted by different bioinformatic missense tools, using functional assay datasets as reference” and Supplementary Table 14 (ST14) “LR towards pathogenicity as predicted by BayesDel across three categories, using an updated functional assay dataset as reference”. Since both ST12 and ST14 set “Moderate evidence of pathogenicity” (Mod P) for BayesDel ≥ 0.3 , following recalibration of the BayesDel thresholds for BRCA1/2, we applied moderate evidence strength for BayesDel PP3 (≥ 0.3). However, ST12 applies “Supporting evidence of benign” (Supp B) for BayesDel < 0.3 and ST14 applies “moderate evidence of benign” (Mod B) for BayesDel ≤ 0.18 . Here, we took a conservative approach and applied “no evidence for BayesDel 0.18-0.3, and applied Supp B for BayesDel ≤ 0.18 . We also applied all other ClinGen VCEP specifications as outlined in ClinGen except that we have now used gnomAD v.4.1

for estimation of population frequency under the BA1 and BS1 specifications because the BRCA1/2 VCEP has recently voted to adopt this strategy.

There are also discrepancies between the main text and Supplementary Table 5, which contains the information on these classifications:

- Variant rarity is incorrectly called PM1 in the main text, but presented as PM2 at supporting level in ST5

Response: We apologize for the inconsistencies. We have corrected the evidence codes throughout the text (Page 4, last paragraph, Page 8 last paragraph) and tables (supplementary Table 6).

Sequence based in silico/splicing predictions is incorrectly called PM3 in the text, and is presented as PP3 in ST5.

Response: We apologize for the inconsistency and have corrected the evidence code throughout the text (Page 4, last paragraph, page 8 last paragraph) and tables (supplementary Table 6).

As noted above, PP3 is also applied at moderate level for sequence-based predictions for missense variants, which differs from the ClinGen BRCA1/2 VCEP guidance.

Response: We applied the ClinGen VCEP specifications listed in "SupplementaryTables_V1.2_2024-11-18.xlsx" ST12 "LR towards pathogenicity as predicted by different bioinformatic missense tools, using functional assay datasets as reference" and ST14 "LR towards pathogenicity as predicted by BayesDel across three categories, using an updated functional assay dataset as reference" which use recalibrated BayesDel thresholds. Because both ST12 and ST14 set moderate evidence for pathogenicity "Mod P" for BayesDel ≥ 0.3 following recalibration of BayesDel for BRCA1/2, we applied moderate evidence strength for BayesDel PP3 (≥ 0.3). However, because ST12 listed BayesDel < 0.3 as Supporting Benign "Supp B" and ST14 listed BayesDel ≤ 0.18 as "Mod B" we took a conservative approach and applied "Supp B" for BayesDel ≤ 0.18 .

- Many additional pieces of evidence are included in ST5 are not described in the main text (PVS1, PS1, PP4, BS2, BP5 codes)

Response: We apologize for this oversight and have now added all of the evidence codes to the main text. (Pages 4 last paragraph and pages 8-9).

- PM5 in the main text is referred to as case-control data. The ClinGen VCEP specifications for BRCA2 state that case-control odds ratio data can be applied under PS4, and that only case-control Likelihood ratio data is preferred and should be under PP4 (or BP5). PM5 is not mentioned in Supplementary Table 5. It is unclear if the case-control data described later in the text was used in the final classification.

Response: We apologize for this error and any confusion caused. We applied multifactorial likelihood clinical data and case-control likelihood ratios based on a recent publication (Zanti et al., 2025) under PP4 (or BP5). We have corrected this issue in the text and header and footnote of supplementary Table 6.

Additional methods descriptions are required for what evidence is used and where it came from (publication, internal data etc), including code strengths drawn from the specifications.

Response: We have thoroughly revised the method section “ClinGen/ACMG/AMP classification” (page 8-9) to describe in detail the evidence used, the respective sources, and the code strengths derived from the specifications. We also added extra columns and detailed footnotes in Supplementary Table 6 to describe the strength of evidence applied and the sources.

Further, to accurately justify the value of functional data for increasing the classification rate, the authors must show the classifications using all other data first, and then the change in classifications by adding the functional data. Figure 2 legend needs to be expanded to be clear that the central pillar highlights the functional weight but that other information was included together with the functional data to reach the final classification shown in the figure. Refer to Supplementary Table 5 in the legend, and this table should be updated to provide more information on the sources of information (as noted above). There appears to be a problem with the bottom of Figure 2 being repeated?

Response: We agree with the reviewer’s point that showing the impact of the added functional data on the classification rate will improve the justification of the value of the functional data. We have now included results with and without the functional data in additional columns “Total ACMG points without PS3/BS3” and “ACMG classification without PS3/BS3” in Supplementary Table 6. We have also added a sentence in the text (page 4, last paragraph) to reflect the increased classification rate “Thus, 91.6% (5849 of 6383) overall and 88.5% (3926 of 4438) of all missense variants were classified as P/LP or B/LB whereas only 12.2% (781 of 6383) of variants overall and 2.9% (127 of 4438) of missense variants were classified without incorporation of the functional results”. We modified Figure 2 by adding a new pillar showing the ACMG classification without incorporation of functional data and by updating the legend for Figure 2 as requested to show that the central pillar reflects only the functional weight. We also corrected the error at the bottom of Figure 2. As noted above, the ACMG classification framework (Supplementary Table 6) has also been updated.

It would also be important to repeat the sensitivity and specificity including functional data at other (lower) strengths. All evidence is useful in variant classification, from supporting to very strong evidence. Lines 92-97 suggest that only Pathogenic_Strong and Benign_Strong/Moderate are useful in classification. In a practical setting, functional results are combined with all other evidence for potential classification - which is what the authors have done in their study, despite saying that only the highest categories have the potential for classification.

Response: We apologize for the misleading sentences in lines 92-97. Indeed, all functional results were combined with other evidence for potential classification in our study regardless of the categories. Importantly, all P categories (strong, moderate and supporting) combined and all B categories (strong, moderate and supporting) combined yielded almost 100% sensitivity and 100% specificity for ClinVar or HDR standards across all models. On that basis it was not possible to distinguish between the models. However, when we took a conservative approach by counting only P_Strong and B_Strong/Moderate category variants, and excluding small numbers of P Moderate, P Supporting and B Supporting variants with lower probabilities of pathogenicity/benignity we were able to distinguish between the models. We have corrected the text accordingly (Page 4, 2nd paragraph)..

Can the authors justify why they used a prior of 0.2. Standard acmg/amp assumes a prior of 0.1 (that was the justification for the bayesian modelling paper), and this prior was carried over for the ClinGen VCEP specifications as well.

Response: We set the prior for the BRCA2 DNA binding region at 0.2 based on AlphaMissense predictions that 22.7% of missense variants in the BRCA2 DBD are likely pathogenic (Huang et al, Nature, 2025). To address the reviewer's comment we also carried out the analysis based on a prior of 0.1. The functional categories based on priors of 0.1 and 0.2 were very similar, with 99.5% concordance among the combined P or combined B and VUS functional categories. We have added this comparison into Supplementary Table 10. We also note that previous publications have also used alternative priors. Specifically, a prior of 0.35 was used for variant analysis in the BRCA1 RING domain (Clark et al, AJHG, 2022), and a prior of 0.0461 and a likelihood ratio for strong evidence of pathogenicity of 33.53 were used in Pejaver et al. AJHG 2022. Because the likelihood ratio cutoff levels depend on the prior, we have derived a table with different priors (0.1, 0.2 and 0.35) and corresponding likelihood ratios, evidence strengths and points for classification (Supplementary Table 9).

It is unclear the spread of classifications for missense variant standards in ClinVar, total 158. Given these data are used to justify the Integrated VarCall model as superior, it is important to know how many were P/LP and how many were B/LB, to ensure sufficient controls on both sides of the equation were included. It should be noted that filtering on missense and ClinVar standards in column K of supplementary Table 5 does not yield 158 variants (the number returned is 125).

Response: We apologize for the confusion. The ClinVar standards used included missense, canonical splice, and intronic variants that were reported in ClinVar and had functional results in all models (nonsense and silent variants were used in the VarCall modeling and were excluded as ClinVar standards for evaluation of performance to avoid circularity). In Supplementary Table 1 we have added four columns (K-N) that list the common ClinVar standards (n=158), the model specific ClinVar standards, the common HDR standards (n=316) and the model specific HDR standards. These correspond to the numbers shown in the panels in Figure 2 and in Supplementary Tables 4 and 5. They also correspond to the standards listed in column J of Supplementary Table 6 (this was column K in Supplementary Table 5 in the original submission), after filtering for missense, canonical and intronic in column F. We have also modified the Text to clarify these numbers and to update the referrals to the various Supplementary Tables.

The OR calculations show good concordance for VarCall results of Pathogenic Strong with both pathogenic missense standards and DBD protein truncating variants. Although the number of variants with Pathogenic Moderate/Supporting VarCall results is quite small, it would be valuable to include those results also in the OR plots. The study would benefit from providing guidance about how to use the Pathogenic Moderate/Supporting and Benign Supporting categories (apply evidence at these levels, or exclude as evidence).

Response: We thank the reviewer for this suggestion. However, due to the limited number of events observed in both cases and controls (<5 in either cases or controls) for Pathogenic Moderate/Supporting and Benign Supporting category variants, we were unable to calculate meaningful ORs for these functional categories. We have added the event counts for each of the Pathogenic and Benign categories to Supplementary Table 8.

Last, the website produced to display the data requires more work to be transparent about what is displayed, and that the information supporting the final classification is downloadable (although with a different table number ST3).

Response: The website (https://brca2-variants-explorer.shinyapps.io/integrated_model/) has been updated to make it easier for those accessing the site to view and/or obtain information. In particular, the data are now downloadable through a specific button link. That table has detailed explanations of the various evidence codes and functional scores for each variant. This corresponds to Supplementary Table 6 in the manuscript. In addition, the website provides a visual representation of the ClinGen/ACMG/AMP classification and the integrated model functional scores for each variant. We believe that viewers will find this NIH supported website very useful.

Further explanation is required in ST5 (and the download file) to explain what the columns mean, and how they were used e.g. spliceR? ; PVS1 point "replace with PS3"?. How were the PS1 points derived? What is the reference variant? Where does the PP4 information come from? Note there are multiple typographical errors in the headers.

Response: We have modified ST5 (now Supplementary Table 6) by adding additional columns indicating the source and reasons for evidence strength and classification points, as well as detailed footnotes to explain each evidence source. The classification process follows the specifications outlined by the ClinGen BRCA1/2 VCEP. Specifically, we have added the SpliceAI score and interpretation of the score in columns N-R and BayesDel in silico prediction scores with assigned thresholds (≤ 0.18 , ≥ 0.3) in column S. PS1 is based on variants located in the same position as known pathogenic variants and/or variants in splicing motifs. Classification Points for PS1 were derived by assigning PS1_Variation Weight based on associated reference variants, which are now listed in columns Z and AA. The PP4/BP5 information is based on multifactorial likelihood ratio data derived from five publications (Parsons et al., 2019, PMC6772163; Easton et al., 2007, PMC2265654; Li et al., 2020, PMC7118020; Goldgar et al., 2004, PMC1182042; Caputo et al., 2021, PMC8546044) and also case-control likelihood ratio data from Zanti et al., 2025, PMC12103537). We have corrected the typographical errors in the headers.

Did the authors use PS3 (functional data) from other studies? Unclear

Response: We did not use any other functional studies for applying PS3. In this manuscript PS3 is based solely on the "Integrated VarCall model" functional data. However, if the "Integrated VarCall model" results are inconsistent with HDR or ClinVar standards, then we labeled the variants as "VUS (discordant)" to be conservative regarding the ACMG classification. (page 5, 1st paragraph).

References

- in addition to reference 1 (from this group), it would be pertinent to refer to the other large scale population-based breast cancer study providing risk estimates for BRCA2 (and indeed a study used to generate odds ratio data).

Response: The other large scale population-based study (Dorling et al NEJM 2021) is reference 3 at the beginning of the Introduction (page 2).

- Check references are appropriate, eg the introduction of the ClinGen BRCA1/2 VCEP (ref 3) appears to refer to a 2019 paper about a multifactorial likelihood model

Response: We have corrected this mistake and put the web link of ClinGen BRCA1/2 VCEP in the text (page 8).

Reviewer #2

Overall, this is a study that will likely be of interest to a broad audience, particularly cancer biologists, geneticists, and clinical geneticists.

Response: We thank the Reviewer for their positive comments about the manuscript.

Major concerns

It was a bit surprising how little information was given on the individual MAVE scoresets other than the single paragraph starting on line 60. A major question I had all of the way through the manuscript is how the integrated models perform relative to the individual input MAVE scoresets. It appears, based on the limited information given, that both sensitivity (89%/93%  98.9% ?) and specificity (~95%/93%  100% ?) are improved, but it is unclear if we can say that for certain. For the integrated scoresets, it is clarified that only Pathogenic_Strong and Benign_Strong/Moderate were used in calculating metrics. Was this the case for the individual MAVE scoresets? I think to alleviate this issue and make it clear to readers that the integrated VarCall method is superior, the individual MAVE scoresets should be included into at least Fig 1. A direct comparison of sensitivity and specificity between the individual MAVE score sets and the integrated models would make it clear how much of an improvement in performance is actually obtained. The authors might further consider supplemental Sankey plots for the individual MAVE scoresets to directly compare with the Integrated VarCall classifications. It might also be interesting to describe in some way, perhaps a supplemental table or Venn diagram, how many and which variants are classified differently by the Integrated VarCall scoreset relative to each input MAVE scoreset. In addition to the average reader of this manuscript, it is not hard to imagine that the appropriate ClinGen VCEP would want this information in order to decide whether to adapt their rulesets to using the Integrated VarCall for assigning PS3/BS3.

Response: We thank the reviewer for this important input and we agree that a head-to-head comparison of all available large datasets from the original MAVE papers and the four new models will provide important additional information. We have now added the two individual MAVE datasets into supplementary tables ST1, 3, 4, 5, Figure 1, and Supplementary figure 1. Variants that changed functional category from the original publications to the Integrated VarCall model are now indicated in ST1 columns "Discrepancy between Huang et al., 2025 and Integrated VarCall model" and "Discrepancy between Sahu et al., 2025 and Integrated VarCall model" and in Supplementary figure 1. We also added text indicating how many variants changed categories from the original publications to the Integrated VarCall model (page 3, last paragraph).

It is not clear to me why only Pathogenic_Strong and Benign_Strong/Moderate variants were used in calculating specificity and sensitivity for each model. I would kindly request the authors to justify this. Unless there is a specific reasoning that was not clear to me from reading this manuscript, it would seem most fair to include all variants scored by each model within each truth set.

Response: We have addressed this comment by adding text as follows "When all P or B category variants (strong, moderate and supporting) from each of the six functional assessment models were compared with ClinVar and HDR control sets, all models yielded close to 100%

sensitivity and specificity. To better distinguish between these models, a more conservative approach that included only variants in Pathogenic_Strong and Benign_Strong/Moderate functional categories, with a higher probability of pathogenic or benign activity, was used to evaluate the performance of each model.” Note that the likelihood ratio for P_Strong based on a prior probability of 0.2 is analogous to a probability of pathogenicity of 73% and variants in the moderate and supporting categories have much lower probabilities of pathogenicity. As such the moderate and supporting categories are more likely to contain error, and are much less convincing as sources of pathogenic and likely pathogenic variants. On this basis, it is important to restrict to the variants with the more convincing probabilities of pathogenicity and benignity for a more accurate comparison of the models.

Minor concerns

There are no accession number(s) given for GEO in the Data Availability...is the data available?

Response: The raw data from each of the published papers that form the basis to the analysis described in this manuscript are provided in GSE270424 and GSE248438). The code and the processed data for the four models described in this manuscript can be found in (najiemayo/Couch_SGE_BRCA2_MAVE_Mayo_NCI_joint_analysis).

In the Fig 2 Sankey plot, it appears that around 20% of the variants that receive an Integrated VarCall score of PS3 strong end up with a final ACMG/AMG classification of VUS. It would be helpful if the authors could comment on this as it seems counterintuitive.

Response: We thank the reviewer for pointing this out. A large proportion of the PS3 strong variants end up with a final ACMG/AMP classification as VUS because of the lack of other evidence needed to get to a minimum of 6 points for classification as likely pathogenic. We have added two sentences in the Discussion to address this: “A substantial number of functionally Pathogenic_Strong variants (219 of 964 (22.7%)) could not be classified due to a lack of information from other sources of evidence. However, because of the strength of the functional evidence many of these variants will likely be classified as P/LP once further evidence becomes available” (page 6, 1st paragraph).

-There seems to be a weird extra segment in the Sankey plot at the bottom of the figure. A portion of the silent variant node appears to have been duplicated? This should be corrected.

Response: We apologize for this mistake. The Sankey plot has been updated.

Reviewer #3:

Reviewer #4:

This work draws on previously published functional assay results and aims to optimise the interpretation of these results through the combination of results from multiple assays. In general, this is a thoughtful and helpful exploration of multiple methods for interpreting this data. My main concerns are how the data generated through the methods described can be incorporated into existing clinical variant interpretation frameworks with appropriate weighting and avoiding circularity.

Please find specific queries and suggestions below:

Suggest to add comment about importance of reclassification of VUS to benign/likely benign (as well as to pathogenic)

Response: We added an additional sentence in the text within the Introduction as follows: “The lack of knowledge about the benign or pathogenic nature of these variants increases uncertainty in patient management and psychological burden for VUS carriers and their families” (Page 2, 1st paragraph of Introduction section). We also extended a sentence in the Discussion as follows “The information provided in this study is expected to substantially improve risk assessment and risk management of individuals found to carry P/LP *BRCA2* DBD variants and will also clarify that many VUS that can now be classified as B/LB are of no relevance to disease” (page 6, 2nd paragraph).

Please revise final sentence “are listed on the ClinVar website (April 2, 2025) along with proposed classification of variants by genetic testing laboratories and over 5000 *BRCA2* VUS and variants with conflicting interpretations.” – doesn’t make sense to me, implies that VUS is not a proposed classification

Response: We have updated this point as follows: “These expert panel-based classifications are listed on the ClinVar website along with proposed classifications of variants by genetic testing laboratories. Overall ClinVar lists over 5000 *BRCA2* VUS and variants with conflicting interpretations.” (page 2, 3rd paragraph).

Additional references needed where VCEP and ACMG/AMP guidance mentioned

Response: We have added references when mentioning VCEP and ACMG/AMP (Richards, S., et al., *Genet Med* 2015; Parsons, et al., *Am J Hum Genet* 2024).

Methods

Unclear why variants with conflicting results were excluded from the concordance model, does the rate of this not provide information? What counts as conflicting?

Response: By definition, discordant variants could not be assigned to a functional category in the concordance model and were excluded. The intent was to evaluate the ability of the concordance model to accurately predict whether variants should be assigned to P or B functional categories, based on consistency between the two MAVE studies, by looking at sensitivity and specificity relative to standard variants. We were not trying to assess how well the concordance model evaluated all variants. That is a very different question. A conflicting or discordant variant is any variant in any P categories in one study that is in a B or VUS category in the other study, and vice versa. The model is now explained in more detail in the Methods section (page 7).

Please explain why only 90% of variants reviewed for the secondary concordance model if all variants looked at for integrated Varcall and GMM model?

Response: The secondary concordance model is based on variants that have concordance between the “integrated VarCall model” and the “integrated GMM model”, which is about 90%. Variants that were discordant between the integrated VarCall and integrated GMM models could not be assigned to a functional category and were excluded from the secondary concordance model. The model is explained in more detail in the Methods section (page 8).

Why was pathogenic_moderate evidence not incorporated into assessment? Could lead to classification if other evidence available (supporting could also, albeit less likely). I can see that

there were fewer variants falling into these categories so was there concern that doing a “banded” analysis of different strength classifications might be underpowered?

Response: To clarify, pathogenic_moderate evidence was used for ClinVar BRCA1/2 VCEP/ACMG based variant classification with each of these variants receiving +2 points under the PS3 specification. Separately assessment of the performance of each model excluded pathogenic_moderate variants. We have addressed this point by adding text (page 4, 2nd paragraph) “When all P or B category variants (strong, moderate and supporting) from each of the six functional assessment models were compared with ClinVar and HDR control sets, all of the models yielded close to 100% sensitivity and specificity. To better distinguish between these models, a more conservative approach that included only variants in Pathogenic_Strong and Benign_Strong/Moderate functional categories, with a higher probability of pathogenic or benign activity, was used to evaluate the performance of each model.” As noted for Reviewer #1, Pathogenic_moderate variants had less than 73% probability of pathogenicity. This likely leads to a higher degree of error in the functional data. By excluding these variants we focus conservatively on the more certain results. This in turn provides a better comparison of the predictive accuracy of the various models. We also considered “banded” analysis of each category but found that the numbers of variants in the pathogenic_moderate and supporting categories were too small for a robust comparison.

Were the pathogenic and benign control variant classifications reached without functional data? If not, the potential limitation of circularity should be mentioned, particularly if involve assays similar to those studied

Response: The pathogenic and benign control variant classifications were extracted from ClinVar (frozen dataset on 5/08/2024) before the two MAVE functional studies were published. We specifically chose this date to avoid circularity based on the MAVE studies. The alternative approach would have been to use current ClinVar submissions but it is clear that many of the more recent submission may be impacted by the MAVE studies. Other previously published functional data may have been used by groups submitting to ClinVar but the impact should have been limited.

On a similar note, although concordance of the model with an alternative (HDR) is notable, it should be made clear that this is not equivalent to clinical classifications incorporating nonfunctional data

Response: We now added a sentence to make clear that the HDR standards are not based on clinical data as follows: “Results were compared to those from the two original studies^{8,9} using established pathogenic and benign clinical standards from ClinVar and functional standards from a well calibrated homology directed DNA repair (HDR) assay.”

Where PM1, PM3 and PM5 codes cited should be PM2, PP3 and PS4 respectively

Response: We apologize for these mistakes. We have made the relevant corrections.

Need to explicitly describe whether overlapping data used between VarCall and the additional data used to produce “ACMG” classifications^{3,4}

Response: There were no overlapping data. The “integrated VarCall model” functional data was based exclusively on functional raw data from the two published MAVE functional studies and

did not include any data from in-silico predictions, other functional studies, allele frequencies, multifactorial likelihood ratios etc.

It would be helpful to explain the VarCall pathogenicity categories earlier, including more detail about how they were calculated in the supplementary methods and how the thresholds were derived, since I can't see how they fit within the ACMG Bayesian model which is often used to quantify other types of evidence

Response: The VarCall model is explained in some detail in the Online Methods section (page 7-8) and in the Github code submission associated with this manuscript, along with the GMM methodology, and was previously described in Huang et al., Nature, 2025. We have added a Supplementary Table 9 to show how the thresholds were derived for each evidence strength level for the VarCall functional data. We also outline in the Introduction and the Results the seven functional categories based on probabilities of pathogenicity or benignity and likelihood ratios of pathogenicity that are consistent with the ClinGen *BRCA1/2* VCEP PS3/BS3 variant classification specifications.

Only a few of the ACMG/AMP codes are mentioned in the online methods – is this deliberate? Please explain how classification was done and by who – i.e. what data was available to the variant classifiers. This last paragraph of online methods also needs rereading for grammar

Response: We apologize for the lack of details for the ACMG/AMP codes. We have now explicitly described all of the applied ACMG/AMP codes in the updated online methods section (page 8-9) and in the footnote of Supplementary Table 6. The ACMG classification was performed by authors Chunling Hu and Megan Bishop, the coordinator of the HBOP VCEP and a member of the *BRCA1/2* VCEP biocurator group.

Please explain justification for using a prior probability of 0.2 for the integrated models

Response: The main result of the Bayesian classification framework by Tavgian et al. 2018 Genet Med, is that the relationship between ACMG/AMP evidence levels (strong, moderate, supporting) follows the Bayesian rule of combining independent evidence: weak evidence can be multiplied to achieve overall strong evidence. Specifically, evidence in a Bayesian framework is modelled in terms of likelihood ratio, $LR = \frac{f(\text{pathogenic}|\text{data})}{f(\text{benign}|\text{data})}$, where $f(\text{pathogenic}|\text{data})$ and $f(\text{benign}|\text{data})$ are the likelihood of being pathogenic or benign for a variant given available data, respectively. The likelihood ratio is referred to as odds of pathogenicity (OddsPath) in Tavgian et al.'s paper. For example, based on the Bayesian classification framework, we have $LR(\text{strong}) = LR(\text{moderate})^2$, $LR(\text{moderate}) = LR(\text{supporting})^2$. At the exponent level, we have $\log_{10}(LR(\text{strong})) = 2 \times \log_{10}(LR(\text{moderate}))$, $\log_{10}(LR(\text{moderate})) = 2 \times \log_{10}(LR(\text{supporting}))$, which matches the combining rules in ACMG/AMP guidelines. These relationships do not depend on the prior probability of a variant being pathogenic. If we denote the prior by p_0 , then the posterior probability of being pathogenic can be calculated as $p_1 = \frac{LR \times p_0}{(LR-1) \times p_0 + 1}$. To define the *LR* cutoff for likely pathogenic, we can set $p_1 = 0.9$ according to the ACMG/AMP definition, and solve $LR(\text{likely pathogenic})$. On the other hand, based on the rule that likely pathogenic can be reached by 1 source of strong evidence and 1 source of moderate evidence, we have $LR(\text{likely pathogenic}) = LR(\text{strong}) \times LR(\text{moderate}) = LR(\text{supporting})^6$. This can be used to set the cutoff for all different evidence levels. Note that the likelihood ratio cutoff levels depend on the prior. For example, in Pejaver et al. AJHG 2022, the prior for a variant is set to 0.0461

and the likelihood ratio for strong evidence is 33.53, which is different from 18.7 odds path for strong evidence used in the Brnich et al, 2019 paper, when assuming a prior 0.1. For reference, we have derived a table with different priors (Supplementary Table 9). We set the prior to 0.2 in the BRCA2 DNA binding region based on the AlphaMissense prediction that 22.7% of missense variants in the BRCA2 DBD were likely pathogenic (Huang et al, Nature, 2025).

Is there a supplementary table 5?

Response: Yes, we do have a supplementary table 5 (now Supplementary Table 6) in excel format, which listed all the variants evaluated in the models with ClinGen BRCA1/2 VCEP and ACMG/AMP code details.

“Subjected to genetic testing” I find a curious phrase to describe clinic testing in the legends of the supplementary tables – consider updating to “in whom genetic testing was undertaken”

Response: We have changed the wording as requested (Supplementary Table 8).

Quite different results for odds ratios using different control datasets – how confident are you that the regression has adjusted sufficiently for ethnicity mismatching between cases and controls? I can't see why the OR would be lower in All of US, if anything if confirmed no cancer phenotype should be higher? If Ambry contained cases with a mixture of ethnicities, unclear why only Europeans from All of Us were selected as controls.

Response: The results using the two different sets of reference controls (AllofUs and gnomADv.4.1) were quite similar. The small differences in odds ratios observed were well within the confidence intervals of the associations and therefore cannot be considered to be different. There were not large differences in the proportions of cases and controls from different ancestral groups so we expect that the adjustment for ancestry in the logistic regression analysis adequately accounted for differences. Note that we now show results from all ancestral populations and not just from Europeans. The bigger differences were seen when comparing the results from the Ambry cases and reference controls with the results from the population-based CARRIERS and BRIDGES cases and age-matched controls. The attenuated results in the population-based studies relative to the high-risk Ambry cases qualifying for testing by NCCN criteria were expected and were consistent with other analyses. Note that the population-based studies were also not adjusted for age effects because these data were not publicly available from the BRIDGES study.

Age matching mentioned in online methods, please present data

Response: Age-matching of cases and controls were used in the CARRIERS and BRIDGES population based studies. The methods were described in detail in each respective publication. We have referenced these two publications in the online methods.

This may be due to the formatting required by the journal but the main text would be more readable if put under sub headings (online methods has clearer structure)

Response: We thank the reviewer for pointing this out. We have now added headings and subheadings for clearer structure of the manuscript.

RESPONSE TO REVIEWERS' COMMENTS

Reviewer #1 (Remarks to the Author):

Comment: The authors have addressed many concerns and issues raised from the initial review.

Response: We thank the reviewer for recognizing our efforts to improve this manuscript.

Comment: Please check sentences are correct after track changes are included, eg
From Abstract: Varcall is mentioned twice in the same sentence, which appears redundant? “An “Integrated VarCall Model” that combined raw functional data from 6383 variants evaluated in both SGE studies in a VarCall probability-based model yielded 98.8% accuracy and out-performed the original SGE studies and three other combined data models”

Response: We mention VarCall twice in the same sentence in the Abstract. One relates to the name of the model “Integrated VarCall Model”, whereas the other is the actual description of the model “in a VarCall probability-based model”. This is not redundancy but we have revised this sentence to avoid any possible confusion as follows: “An “Integrated VarCall Model” that combined raw functional data from 6383 variants evaluated in both SGE studies yielded 98.8% accuracy and out-performed the original SGE studies and three other combined data models...”

Comment: From Discussion: An “Integrated VarCall model” that combined raw functional scores from the two studies of substantially outperformed a simple “Concordance model”, the individual SGE studies, and other integrated models for improved functional interpretation and ClinGen/ACMG/AMP classification of BRCA2 variants.

Response: We have deleted the “of” from this sentence.

Comment: While the ST12 and ST14 show LRs that fall into the Moderate evidence of pathogenicity, there is only reference to PP3 code being applied in all other places in the specifications. On the ClinGen Specifications website there is no allowance described for modification of PP3 to other evidence categories. Additionally, the recent publication by the VCEP (PMID: 39142283) states in the PP3 section: “The VCEP opted, conservatively, to apply this evidence type at supporting weight only (Table 2).” Filtering on ST6, this will only change classification for 24 variants that are LP with 6 points, including 2 points from PP3. It is particularly important to use the conservative application of PP3/BS4 for this paper specifically, since the bioinformatic tools were calibrated against functional data results, and there will be circularity due to correlation of the MAVE results with the original calibration reference set. [Some might argue, validly, that PP3/BP4 should not be used at all]

Response: We have now applied the conservative “supporting” weight of evidence for PP3/BP4. All related text, methods, tables, and figures now reflect this change.

Comment: Please note the change to using gnomAD v 4.1 specifically in the methods section for ClinGen/ACMG/AMP classification, to avoid confusion for readers who refer to the specifications version as stated in the paper

Response: We have modified the methods section to reflect this as follows: “PM2_Supporting based on absence from gnomAD v4.1 non-UKBB (this is updated from gnomAD v2.1 and v3.1 that were listed as the preferred datasets in the *BRCA1/2* VCEP specifications)”

Comment: This section (“ClinGen/ACMG/AMP classification” (page 8-9)) is greatly improved. There are 2x PS3 statements in a row. Does this statement refer to PS1 (Splicing) rather than PS3? “PS3), applied per ClinGen VCEP specifications table 5 and appendix table 17”.

Response: We apologize for this error and have corrected it.

Comment: For Supplementary Table 6, please add a marker to the footnote for ACMG/AMP criteria (as is done for HDR and ClinVar standards. Potentially separate ones for each code. E.g. it was not clear that the reference numbers for LR sources were contained within description for PP4/BP5. There is no indication of the sources for application of RNA codes in the Supplementary Table or the manuscript text. There is a typographical error in header for column AT, and the header needs more explanation – what does before “_” refer to?

Response: We have added a marker to the footnote for ACMG/AMP criteria. In the footnote of Supplementary Table 6 we now list the sources of the RNA codes “PVS1[RNA]: intronic/silent/missense; per ClinGen *BRCA1/2* VCEP specifications Figure 1B; Appendix D Figure 9; Appendix E Table 9; Supplementary Table 2 and 3; replace bioinformatic codes with appropriate PVS1 [RNA] code.” ClinGen *BRCA1/2* VCEP specification Supplementary Tables 2 and 3 have detailed Splicing Assay results to inform the RNA codes applied.

Comment: The addition of pillar 1 (to Figure 2) showing initial classification highlights the difference adding in functional data can make. The impact is less obvious by going through the variant type pillar. It looks like the figure is now addressing two different points, 1) how likely variants of a certain type are to get certain functional assay outcomes and their subsequent classification, 2) how classification changes with the addition of these functional data from this study. If the main purpose of the figure is to show 2), reclassification impact of these functional data, consider removing variant type (which is available in ST6).

Response: We agree with the reviewer’s comments and have removed the variant-type pillar as requested.

Comment: Suggest adding a comment on this (analysis and classification restricted to MAVE functional data) in the discussion as a limitation of the study. While the two MAVE publications incorporated in this study represent a large proportion of published functional assay results in *BRCA2*, previous work has

been published in this area and may contribute to variant classification as well (i.e. not all changed classifications are new when incorporating other functional assay work).

Response: We have added a new sentence in the Discussion as follows: “Furthermore, while the two MAVE publications incorporated in this study represent a large proportion of published functional assay results in *BRCA2*, previous smaller scale functional studies of *BRCA2* variants have been published could also potentially contribute to variant classification”.

Comment: The website is improved, but requires more editing to distinguish between what is a functional assay “call” and strength of evidence versus final classification using ACMG/AMP guidelines. (1) It is not clear if the colored boxes refer to final classification. This is important since VUS is also used as a category for the functional code assignment. (2) “Variants were classified based on the functional scores obtained from the Integrated VarCall Model” should rather read something like “were assigned an ACMG/AMP functional code and strength based on functional scores from the VarCall Model” (3) For ACMG/AMP classification, the functional evidence was combined with other available evidence, based on ClinGen ENIGMA BRCA1 and BRCA2 VCEP Specifications - and please provide the specifications version. For the data field description, clarify that the ACMG/AMP classification was following ClinGen VCEP specifications for this gene (and version number). (4) The column header Integrated Model Classification should be changed to something like “Functional code assigned based on Integrated Model” or “Integrated Model Category”.

Response: We have made substantial changes to the Variant Explorer site in response to the reviewer’s suggestions. (1) We have clarified that the colored boxes refer to ACMG/AMP classification by adding a line just above the box plots stating "Choose an exon or enter a genomic position/amino acid number to explore the results (box colors refer to ACMG/AMP classification of the variants)". (2) We have modified the text as follows "Variants were assigned an ACMG/AMP functional code and strength based on functional scores from the Integrated VarCall Model". (3) We have changed the label as follows: "For ACMG/AMP classification, the functional evidence was combined with other available evidence, based on ClinGen ENIGMA BRCA1 and BRCA2 VCEP Specifications (Version: 1.2.0) according to Parsons et al., 2025 (PMID: 39142283)". (4) We have changed the column header from "Integrated Model Classification" to “Integrated Model Category”.

Reviewer #2 (Remarks on code availability):

Review of code is outside the expertise of the reviewers.

Reviewer #3 (Remarks to the Author):

The authors addressed all of my concerns in their revised manuscript.

Reviewer #4 (Remarks to the Author):

Reviewer #4 (Remarks to the Author):

The authors have satisfactorily addressed my comments and the manuscript is much clearer.

Comment: They have explained why a prior of 0.2 was used for the DNA binding domain in their rebuttal, but I would recommend they include this rationale in the main text of the manuscript.

Response: We thank the reviewers for their positive comments. We have included the rationale for using a prior of 0.2 in the main text as follows: "A prior probability of pathogenicity of 0.2 for variants in the DNA binding region was used based on AlphaMissense predictions that 22.7% of missense variants in the BRCA2 DBD are pathogenic."

RESPONSE TO REVIEWERS' COMMENTS

Reviewer #1 (Remarks to the Author):

Reviewer comment: In the supplemental table ST6 (632986_2_data_set_12263831_t90cvc.xlsx), 2 points are still applied for PP3 based on BayesDel predictions. This is carried into the sum of points and may impact the overall classification numbers. Filtering in ST6 matches the current numbers in the manuscript, which means that this change hasn't been taken through all of the data, although statements are updated in the text.

Response: We apologize for overlooking the contents of Supplementary Table ST6 when we downscaled the PP3 *in silico* prediction strength of evidence from pathogenic moderate (+2 points) to pathogenic supporting (+1 points). We have corrected the Table and now show all of the PP3 entries based on BayesDel predictions as +1 point. This correction results in reclassification of a small number of variants. All relevant texts, tables and figures have been modified to account for these changes in classification.